# Deciphering Microbiota of Acute Upper Respiratory Infections: A Comparative Analysis of PCR and mNGS Methods for Lower Respiratory Trafficking Potential

Sadia Almas [1], Rob E. Carpenter [1,2,*], Anuradha Singh [3], Chase Rowan [1], Vaibhav K. Tamrakar [3,4] and Rahul Sharma [1,3]

1   Department of Research, Advanta Genetics, 10935 CR 159, Tyler, TX 75703, USA
2   Department of Human Resource Development, University of Texas at Tyler, 3900 University Boulevard, Tyler, TX 75799, USA
3   ICMR-National Institute of Research in Tribal Health, Jabalpur 482003, India
4   RetroBioTech LLC, 838 Dalmalley Ln, Coppell, TX 75019, USA
*   Correspondence: rcarpenter@uttyler.edu; Tel.: +1-903-530-1700

**Highlights:**

**What are the main findings?**

- Although there was a high concordance between methodologies, a hybridization-capture-based mNGS workflow was able to detect 29 additional upper respiratory microorganisms versus PCR.
- The identified microorganisms were rapidly characterized into three phenotypic groups for infectivity and trafficking potential.

**What is the implication of the main finding?**

- A hybridization-capture-based mNGS workflow can provide a comprehensive yet clinically relevant microbiology profile of acute upper respiratory infection.
- Deciphering upper respiratory microbiota with phenotypic grouping has potential to provide respiratory medicine a tool to better manage immunocompromised, immunocompetent with comorbidity and complex respiratory cases.

**Abstract:** Although it is clinically important for acute respiratory tract (co)infections to have a rapid and accurate diagnosis, it is critical that respiratory medicine understands the advantages of current laboratory methods. In this study, we tested nasopharyngeal samples (*n* = 29) with a commercially available PCR assay and compared the results with those of a hybridization-capture-based mNGS workflow. Detection criteria for positive PCR samples was Ct < 35 and for mNGS samples it was >40% target coverage, median depth of 1X and RPKM > 10. A high degree of concordance (98.33% PPA and 100% NPA) was recorded. However, mNGS yielded positively 29 additional microorganisms (23 bacteria, 4 viruses, and 2 fungi) beyond PCR. We then characterized the microorganisms of each method into three phenotypic categories using the IDbyDNA Explify® Platform (Illumina® Inc, San Diego, CA, USA) for consideration of infectivity and trafficking potential to the lower respiratory region. The findings are significant for providing a comprehensive yet clinically relevant microbiology profile of acute upper respiratory infection, especially important in immunocompromised or immunocompetent with comorbidity respiratory cases or where traditional syndromic approaches fail to identify pathogenicity. Accordingly, this technology can be used to supplement current syndrome-based tests, and data can quickly and effectively be phenotypically characterized for trafficking potential, clinical (co)infection, and comorbid consideration—with promise to reduce morbidity and mortality.

**Keywords:** metagenomics; respiratory infections; phenotype; hybridization approach; immunocompetent; laboratory; antimicrobial resistance; bacteria; viruses; fungus

## 1. Introduction

Preceding or concurrent upper respiratory (co)infections can have harmful trafficking effects on lower respiratory disease. Lower respiratory (co)infections are a communal source of worldwide morbidity and mortality [1]. The clinical picture of human upper and lower respiratory infections can be complex and heterogenous since etiological agents (i.e., bacteria, fungi, viruses, and parasites) can be present alone or in combination. For example, the consequences of viral–bacterial (co)infections have been increasingly recognized for affecting the manifestation and prognosis of community-acquired pneumonia and can profoundly impact concomitant development of respiratory disease, frequently resulting in the need for intensive care [2–6]. Especially at risk are children under the age of one, pregnant women, the elderly, and immunocompromised hosts. Immunocompetent individuals with comorbid illness are also at increased risk of severe respiratory infection often requiring intensive care [7]. The recent COVID-19 pandemic further highlighted that viral performance often has devastating effects on human health when coupled with fungal and bacterial (co)infections [8]. No doubt, the burden of respiratory (co)infection is a major threat to global health, and the need for timely and accurate diagnosis is universal [9,10].

It Is clinically important for acute respiratory tract (co)infections to have a rapid and accurate diagnosis to reduce the risk of protracted (co)infections and advance the application of pathogen-specific medication—taking into account the worsening universal problem of antibiotic-resistant microbes [11,12]. For example, multiplex polymerase chain reaction (PCR) assays have advanced diagnosis for numerous respiratory pathogens and antimicrobial resistance (AMR) markers in a single panel, reducing diagnosis time and bypassing other serial tests like serology and culture to identify respiratory (co)infections [13]. However, it is critical to understand these syndromic approaches have limitations rooted in techniques based upon a priori assumptions and a validated scope of target-specific agents; meaning these methods are biased to a set of predetermined microbes with limited capacity to discover or differentiate clinically relevant strains or genotypes [9,14]. As a result, these traditional laboratory approaches may not identify (co)infectivity, trafficking potential, fastidious microorganisms, rare and atypical pathogens, or agents inviable on culture after antimicrobial therapy.

On the other end of the spectrum, the target agnostic approach of metagenomic next-generation sequencing (mNGS) technology has potential to accurately identify respiratory (co)infections without a priori knowledge aimed at broadening pathogen discovery, shortening detection time for certain microorganisms, and with detection strength less affected by past antibiotic exposure. This hypothesis-free approach has emerged as a promising laboratory method; yielding higher pathogen identification, uncovering progressive (co)infections, and for directly influencing patient care—including suitable antibiotic coverage and reduced mechanical ventilation [15–17]. One of the greatest attributes of mNGS may be its ability to capture a patient's microbiome to detect comorbid infections that may complicate treatment and recovery [18]. However, mNGS, too, has limitations that are critical to understand for respiratory medicine. Because of the target agnostic approach, mNGS data have potential to query the whole microbiome in the sample; the significance in acquiring thousands to millions of short DNA sequences from a single respiratory sample can be a taxonomic burden and relies primarily on being able to resolve clinically relevant data. In doing so, several challenges exist in experimental design and computational analysis between pathogenicity and the true microbiome in samples such as respiratory fluid [19]—with interpretation of the data for case relevance in respiratory medicine another matter altogether.

For these reasons, targeted (precision) metagenomics—a hybridization-capture-based mNGS approach—is becoming more commonly considered for targeted sequencing in clinical settings [20–23]. Hybridization capture is especially helpful because it uses biotinylated oligonucleotide probes to focus on specific genomic regions of interest. Accordingly, its multiplexing capacity is improved by enabling "molecular barcodes" that can be ligated, combined, and pooled with several samples at equal mass, reducing workflow effort and

cost. Moreover, a targeted hybridization-capture-based mNGS approach can maximize on-target reads, improve mutation exactitude, and provide superior performance with complex sequences, making it an especially appealing option for analyzing respiratory infections. And although researchers have demonstrated that such hybridized workflows can provide critical pathogen-specific sequences for lower respiratory (co)infections [24], preceding or concurrent consideration of upper respiratory (co)infections with potential to orchestrate specific leukocyte trafficking molecules to inform a course of lower respiratory pathology remains an open challenge in the fields of laboratory and respiratory medicine. Pathogen-specific and excessive leukocyte recruitment and activation may lead to life-threatening manifestations of the disease in lower respiratory track. Therefore, pathogen-specific treatment guided from accurate identification has potential to help maintain the immune response homeostasis [25].

The purpose of this study is twofold. First, this research helps extend our understanding of the potential clinical utility of a hybridization-capture-based mNGS workflow for respiratory medicine by deploying a Respiratory Pathogen ID/AMR enrichment panel (RPIP) (187 bacteria, 42 viruses, 53 fungi, and 1218 AMR; see https://www.illumina.com/products/by-type/sequencing-kits/library-prep-kits/respiratory-pathogen-id-panel.html (accessed on 12 December 2022) for additional information) to test nasopharyngeal samples (*n* = 29) of individuals suspected of acute upper respiratory infection and comparatively analyze the results of identical samples using the Fast Track Diagnostic (FTD®) Respiratory Pathogens (Siemens Healthineers, Erlangen, Germany) 33 (RUO) PCR panel (21 viruses, 12 bacteria, 1 fungus, and 2 AMR). Second, we characterized the discovered microorganisms of each method into phenotypic categories using the IDbyDNA Explify® Platform for consideration of trafficking concerns to the lower respiratory region for the attention of respiratory medicine.

## 2. Materials and Methods

### 2.1. Nucleic Acid Isolation

Nucleic acid was extracted from nasopharyngeal samples (*n* = 29) collected from individuals recommended for respiratory pathogen PCR testing at Advanta Genetics (Tyler, TX, USA) from January 2022 to June 2022. Two different methods were used for nucleic acid extraction for PCR and mNGS analysis; total nucleic acid was extracted for PCR analysis, and DNA and RNA were extracted separately for mNGS analysis. Briefly, total nucleic acid isolation was performed as a part of routine diagnostic testing using the MagNA Pure 96 (MP96) DNA and Viral NA Small Volume Kit (Cat # 06543588001; Roche Diagnostics GmbH, Mannheim, Germany). Briefly, samples were lysed with 340 µL of lysis buffer and 10 µL of proteinase K (Invitrogen Cat # 4333793; Thermo Fisher Scientific, Waltham, ME, USA) at 55 °C for 10 min, followed by extraction via the MP96 instrument. Extracted nucleic acids were stored at −80 °C until used for PCR testing.

DNA and RNA from each sample were extracted separately using the Zymo research reagents as per the protocol provided in the Explify Respiratory Pathogen ID/AMR Panel User Guide (https://www.illumina.com/content/dam/illumina-support/documents/documentation/software_documentation/idbydna/CUS-USRG_9001-03-Explify-Enrichment-Respiratory-Pathogen-ID_AMR-Panel-Sample-Processing-and-Sequencing-Library-Preparation-User-Guide-220418.pdf accessed on 1 February 2023). Each sample was spiked with T7 bacteriophage DNA (Microbiologics, St. Cloud, MN, USA), delivering a final concentration of $1.2 \times 10^7$ plaque forming units (PFU/mL) of sample. Copies of T7 were used for computing the absolute concentration of the target copies detected in the samples. Briefly, 400 µL of the sample was homogenized and lysed by vertexing in ZR Bashing Beads (Cat# S6012-50; Zymo Research, Irvine, CA, USA). Homogenate supernatant was mixed with DNA/RNA Lysis Buffer (Cat # D7001-1; Zymo Research, Irvine, CA, USA), and DNA was first extracted using a Spin-Away Filter (Cat #. C1006; Zymo Research, Irvine, CA, USA). Flow-through from the DNA extraction was used for RNA extraction using Zymo-Spin IIC (Cat #. C1011; Zymo Research, Irvine, CA, USA). RNA was treated with the Zymo DNases I enzyme. (Cat # E1009-A; Zymo Research, Irvine, CA, USA).

## 2.2. PCR Testing Using Fast Track Diagnostic® Assay

Real-time PCR testing was performed using a TaqMan chemistry-based Fast Track Diagnostic® (FTD) Respiratory Pathogens 33 (RUO) kit (Siemens Healthineers, Erlangen, Germany). Ten microliter multiplex reactions targeting 13 bacteria, 19 viruses, and one fungal pathogen were performed in a 384-well plate format on a Light Cycler® 480 System(Roche) instrument (Supplementary Table S1). One-step reverse transcription PCR was performed in 11 multiplex PCR reactions, and each reaction was targeted to detect three pathogens. The multiplex real-time RT-PCR thermal cycling profile for the FTD kit (Siemens Healthineers, Erlangen, Germany) included cDNA synthesis at 50 °C for 15 min, and initial denaturation at 95 °C for 10 min followed by 40 cycles of PCR amplification at 95 °C for 8 s and 60 °C for 34 s. PCR results were considered positive for the targets if the threshold cycle (Ct) values were ≤35 paired with sigmoidal amplification curves.

## 2.3. Library Preparation and Enrichment

Sequencing libraries were prepared for the mNGS using Illumina®/IDbyDNA Respiratory Pathogen ID/AMR Panel (RPIP) protocol and reagents (Illumina® Inc, San Diego, CA, USA). Briefly, cDNA was prepared from the RNA and combined with the DNA in equal volumes. Libraries were constructed by DNA tagmentation and adapter ligation using an Illumina® RNA Prep with enrichment kit (Illumina® Inc, San Diego, CA, USA). Libraries were enriched for the microbial content by hybridization with the RPIP probes for 2 h. Captured libraries were amplified for 14 cycles and cleaned using AmPure XP beads (Beckman Coulter, Brea, CA, USA). Two NATtrolTM Respiratory Panel 2.1 (RP2.1) Controls (Cat# NATRPC2.1-BIO) (ZeptoMetrix, Buffalo, NY, USA) and a blank viral transport medium (VTM) (Criterion Chemistries, Pelham, AL, USA) were included as positive and negative controls, respectively, with each batch of library preparation and sequencing. Libraries were quantified using a Qubit Flex Fluorometer (Thermo Fisher Scientific, Waltham, Main, USA), and fragment sizes of representative libraries were analyzed in an Agilent 5200 Fragment Analyzer. The enriched libraries were then pooled to an equimolar concentration and normalized to 1 nM concentration. The final library pool was denatured and neutralized with 0.1 $n$ NaOH and 200 mM Tris-HCL (pH-8). The denatured libraries were further diluted to a loading concentration of 2 pM. Dual indexed paired-end sequencing with 75 bp read length was performed using a high-output flow cell (150 cycles) on an Illumina MiniSeq® instrument (Illumina® Inc, San Diego, CA, USA). Although the goal depth for this workflow was 1.0 million reads, samples with 0.5 million reads were included in the downstream analysis.

## 2.4. Bioinformatic Analysis Using the Explify® Platform

Sequencing data were analyzed using an automated IDbyDNA Explify® Platform data analysis solution (v1.0.1). This software detects 282 pathogens, covering more than 95% of common and rare pathogens of respiratory infections. This software also detects 1218 AMR markers to predict the resistance of 12 common bacterial pathogens to 16 commonly used antibiotics. However, AMR analysis was not included in this comparative study because the PCR panel used in the study was limited to microorganism detection and did not include AMR marker assays. Moreover, respiratory samples were also not tested by microbiological culture sensitivity.

Following analysis, a report was generated that included a detailed text-based (JSON format) and .pdf document that contained the quantitative identification of viruses, bacteria, and fungi in each sample, including the AMR markers. Each identified microorganism was assigned to phenotypic groups (1, 2, and 3) based on its potential pathogenic status. Group 1 microorganisms are frequently considered part of the normal flora but may be associated with disease in certain settings. Microorganisms in group 2 are frequently associated with the disease, and group 3 microorganisms are generally considered to be associated with the disease. Accurate detection of the known microorganisms in positive controls was used for defining the acceptance criteria for target detection in clinical samples.

## 3. Results

Identical nasopharyngeal samples (*n* = 29) collected from individuals with suspected acute upper respiratory infections were tested using a syndromic PCR panel (Supplementary Table S1) and a hybridization-capture-based mNGS workflow (Supplementary Table S2). All data were analyzed qualitatively. Serial dilutions of Zeprometrix respiratory controls (1 and 2) were tested with each batch of the clinical samples, and only controls with >0.5 million total reads were considered for accurate detection of all included targets. Thus, samples without the minimum of 0.5 million reads were excluded from further analysis. Further, the minimum coverage and reads per kilobase per million reads mapped (RPKM) required for >90% accurate detection of the targets in controls was used as acceptance criteria for microorganism detection in clinical sample results. Detection criteria for positive microorganisms were Ct < 35 in the PCR assay, >40% target coverage, median depth of $1\times$, and RPKM > 10 in the mNGS assay. The results from the two methodologies were comparatively analyzed and phenotypically characterized into groups according to their infectivity potential.

### 3.1. Bioinformatic Analysis Using the Explify® Platform

Analyte-specific PCR analysis detected 28/29 samples positive for one or more microorganism(s). One sample was negative for all tested microorganisms. Overall, 15 etiological agents (4 bacteria and 11 viruses) were identified among the 28 positive samples tested by PCR (Table 1). Analysis of the 28 positive samples revealed 9 samples exclusively positive for bacteria and 9 samples exclusively positive for viruses, whereas 10 samples were concomitantly positive for both bacteria and viruses. The most common microorganisms detected by PCR were *Moraxella catarrhalis*, *Hemophilus influenza*, and *Streptococcus pneumoniae*.

**Table 1.** Fast Track Diagnostic® (FTD) assay results (*n* = 29).

| FTD® PCR Panel | Microorganism Classification | PCR Positive |
|---|---|---|
| *Moraxella catarrhalis* | Bacteria | 13 (45%) |
| *Haemophilus influenzae* | Bacteria | 9 (31%) |
| *Streptococcus pneumoniae* | Bacteria | 9 (31%) |
| *Staphylococcus aureus* | Bacteria | 5 (17%) |
| Human rhinovirus | Virus | 4 (14%) |
| Influenza A virus (H3N2) | Virus | 4 (14%) |
| Human coronavirus OC43 | Virus | 3 (10%) |
| Human metapneumoviruses | Virus | 3 (10%) |
| Human respiratory syncytial viruses | Virus | 3 (10%) |
| Human parainfluenza 1 virus | Virus | 2 (7%) |
| Enterovirus | Virus | 1 (3%) |
| Human parainfluenza 2 virus | Virus | 1 (3%) |
| Human parainfluenza 3 virus | Virus | 1 (3%) |
| Human parainfluenza 4 virus | Virus | 1 (3%) |

The phenotypic grouping of positive microorganisms was then classified according to their infectivity potential: 0/28 (0%) microorganisms were phenotypically classified (phenotypic group 1) as part of the normal flora, colonizers, or contaminants; 3/28 (11%) microorganisms were phenotypically classified (phenotypic group 2) as frequently associated with respiratory disease; and 12/28 (43%) microorganisms were phenotypically classified (phenotypic group 3) as pathogenic for respiratory disease.

### 3.2. Bioinformatic Analysis Using Hybridization-Capture-Based mNGS Workflow

The hybridization-capture-based mNGS workflow used in this study probed an additional 249 microorganisms (23 viruses, 174 bacteria, and 52 fungi) that were not included in the PCR panel. However, only samples yielding >0.5 million reads were included in the downstream mNGS analysis; microorganisms with coverage $\geq 40.00\%$, median depth $\geq 1x$,

and RPKM $\geq 10.00$ were considered positive. The hybridization-capture-based mNGS workflow identified 44 microorganisms (27 bacteria, 2 fungi, and 15 viruses) in 29 samples (Table 2; Figure 1).

**Table 2.** Illumina® RPIP mNGS panel positive microorganisms.

| Illumina® RPIP mNGS Panel | Microorganism Classification | mNGS Positive |
|---|---|---|
| *Moraxella catarrhalis* | Bacteria | 12 (41%) |
| *Dolosigranulum pigrum* | Bacteria | 12 (41%) |
| *Haemophilus influenzae* | Bacteria | 9 (31%) |
| *Streptococcus pneumoniae* | Bacteria | 9 (31%) |
| *Stenotrophomonas maltophilia* | Bacteria | 7 (24%) |
| *Pseudomonas aeruginosa* | Bacteria | 6 (21%) |
| *Staphylococcus aureus* | Bacteria | 5 (17%) |
| *Corynebacterium pseudodiphtheriticum* | Bacteria | 4 (14%) |
| Human rhinovirus A | Virus | 4 (14%) |
| Influenza A virus (H3N2) | Virus | 4 (14%) |
| *Corynebacterium propinquum* | Bacteria | 3 (10%) |
| Human coronavirus OC43 | Virus | 3 (10%) |
| Human metapneumovirus | Virus | 3 (10%) |
| *Ochrobactrum anthropi* | Bacteria | 3 (10%) |
| *Prevotella melaninogenica* | Bacteria | 3 (10%) |
| Respiratory syncytial virus B | Virus | 3 (10%) |
| *Rothia mucilaginosa* | Bacteria | 3 (10%) |
| *Streptococcus mitis* | Bacteria | 3 (10%) |
| *Actinomyces graevenitzii* | Bacteria | 2 (7%) |
| *Alternaria alternata* | Fungus | 2 (7%) |
| *Campylobacter concisus* | Bacteria | 2 (7%) |
| *Capnocytophaga leadbetteri* | Bacteria | 2 (7%) |
| Cytomegalovirus (CMV) | Virus | 2 (7%) |
| *Gemella haemolysans* | Bacteria | 2 (7%) |
| Human parainfluenza virus 1 | Virus | 2 (7%) |
| SARS-CoV-2 (2019-nCoV) | Virus | 2 (7%) |
| *Veillonella parvula* | Bacteria | 2 (7%) |
| *Achromobacter xylosoxidans* | Fungus | 1 (3%) |
| *Actinomyces naeslundii* | Fungus | 1 (3%) |
| Coxsackievirus A | Virus | 1 (3%) |
| Enterovirus D68 | Virus | 1 (3%) |
| *Fusarium proliferatum* | Fungus | 1 (3%) |
| *Fusobacterium necrophorum* | Fungus | 1 (3%) |
| *Haemophilus haemolyticus* | Bacteria | 1 (3%) |
| *Haemophilus parainfluenzae* | Bacteria | 1 (3%) |
| Human parainfluenza 4 virus | Virus | 1 (3%) |
| Human parainfluenza virus 2 | Virus | 1 (3%) |
| Human parainfluenza virus 3 | Virus | 1 (3%) |
| Human rhinovirus C | Virus | 1 (3%) |
| Influenza A virus (H1N1) | Virus | 1 (3%) |
| *Neisseria flavescens* | Bacteria | 1 (3%) |
| *Neisseria lactamica* | Bacteria | 1 (3%) |
| *Pseudomonas stutzeri* | Bacteria | 1 (3%) |
| *Streptococcus intermedius* | Bacteria | 1 (3%) |

The results demonstrated that only three samples were positive for a single organism (*S. aureus*, HRV-C, A/H1N1), five samples were exclusively co-infected with bacterial species, and two samples were positive for a multi-viral infection. Bacterial and viral coexistence was the most common presentation identified ($n = 16$), followed by bacterial and fungal coexistence in two samples. Only one sample was found concurrently colonized with bacteria, viruses, and fungus. The phenotypic grouping of positive microorganisms was then classified according to their infectivity potential: 14/44 (31%) microorganisms

were phenotypically classified (phenotypic group 1) as part of the normal flora, colonizers, or contaminants; 15/44 (34%) microorganisms were phenotypically classified (phenotypic group 2) as frequently associated with respiratory disease; and 15/44 (34%) microorganisms were phenotypically classified (phenotypic group 3) as pathogenic for respiratory disease.

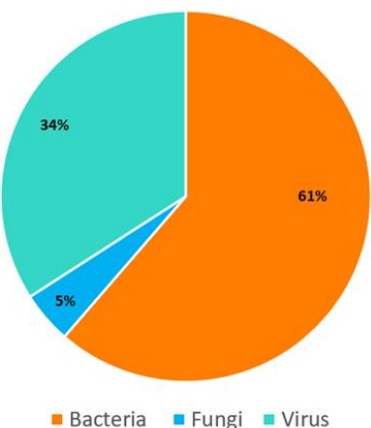

**Figure 1.** Classification of microorganisms detected by mNGS (coverage $\geq$ 40.00%, median depth $\geq 1\times$ and RPKM $\geq$ 10.00). To avoid any errors during position changes, please provide the combined image instead of editable piece in the figure.

### 3.3. Comparative Analysis between PCR and mNGS Assays

The results of both assays were analyzed, compared, and phenotypically classified (Table 3). A high degree of concordance (98.33% PPA and 100% NPA) was recorded between the PCR and mNGS results for the targets (*n* = 33) shared by both panels. Only one sample showed discordance when *M. catarrhalis* was detected by PCR and not mNGS. While 4 bacteria and 11 viruses were concurrently detected by the PCR and mNGS panels, mNGS yielded positive for 29 additional microorganisms (23 bacteria, 4 viruses, and 2 fungi) beyond the PCR panel. Granted, the mNGS panel identified the colonization of a wide range of upper respiratory tract flora with (14/44; 13 bacteria, 1 fungus) that were unlikely pathogenic and classified to phenotypic group 1. Among the 15 microorganisms detected and assigned to phenotypic group 2, mNGS analysis exclusively detected 12/15—these microorganisms are frequently associated with lower respiratory disease, and detection of these microorganisms is potentially clinically significant. Coxsackievirus A, SARS-CoV-2 (2019-nCoV), and HRV-C were exclusively identified by mNGS and classified to phenotypic group 3 as pathogenic with Explify® (Illumina® Inc, San Diego, CA, USA) analysis. Two samples found positive for SARS-CoV-2 and were separately tested with SARS-CoV-2-specific PCR assays; both were confirmed positive. The remaining 12 organisms assigned to phenotypic group 3 (generally considered disease-associated) were in parallel with the PCR panel and were concurrently detected by both assays (Figure 2).

**Table 3.** PCR vs mNGS phenotypic grouping of upper respiratory tract microorganisms.

| **Phenotypic Group 1** | | |
| --- | --- | --- |
| **No.** | **PCR** | **mNGS** |
| 1 | | *Haemophilus haemolyticus* |
| 2 | | *Neisseria flavescens* |
| 3 | | *Neisseria lactamica* |
| 4 | | *Dolosigranulum pigrum* |
| 5 | | *Alternaria alternata* |
| 6 | | *Campylobacter concisus* |
| 7 | | *Capnocytophaga leadbetteri* |
| 8 | | *Gemella haemolysans* |
| 9 | | *Veillonella parvula* |
| 10 | | *Corynebacterium propinquum* |

**Table 3.** *Cont.*

| No. | PCR | mNGS |
|---|---|---|
| **Phenotypic Group 1** | | |
| 11 | | *Ochrobactrum anthropi* |
| 12 | | *Prevotella melaninogenica* |
| 13 | | *Rothia mucilaginosa* |
| 14 | | *Corynebacterium pseudodiphtheriticum* |
| **Phenotypic Group 2** | | |
| 1 | *Moraxella catarrhalis* | *Moraxella catarrhalis* |
| 2 | *Haemophilus influenzae* | *Haemophilus influenzae* |
| 3 | *Streptococcus pneumoniae* | *Streptococcus pneumoniae* |
| 4 | | Achromobacter xylosoxidans |
| 5 | | *Actinomyces naeslundii* |
| 6 | | *Fusobacterium necrophorum* |
| 7 | | *Haemophilus parainfluenzae* |
| 8 | | *Pseudomonas stutzeri* |
| 9 | | *Streptococcus intermedius* |
| 10 | | *Actinomyces graevenitzii* |
| 11 | | *Cytomegalovirus (CMV)* |
| 12 | | *Fusarium proliferatum* |
| 13 | | *Streptococcus mitis* |
| 14 | | *Pseudomonas aeruginosa* |
| 15 | | *Stenotrophomonas maltophilia* |
| **Phenotypic Group 3** | | |
| 1 | Enterovirus | Enterovirus |
| 2 | Human parainfluenza virus 2 | Human parainfluenza virus 2 |
| 3 | Human parainfluenza virus 3 | Human parainfluenza virus 3 |
| 4 | Human parainfluenza virus 4 | Human parainfluenza virus 4 |
| 5 | Influenza A virus (H1N1) swl | Influenza A virus (H1N1) swl |
| 6 | Human parainfluenza virus 1 | Human parainfluenza virus 1 |
| 7 | Human coronavirus OC43 | Human coronavirus OC43 |
| 8 | Human metapneumovirus | Human metapneumovirus |
| 9 | Respiratory syncytial virus B | Respiratory syncytial virus B |
| 10 | Human rhinovirus A | Human rhinovirus A |
| 11 | Influenza A virus (H3N2) | Influenza A virus (H3N2) |
| 12 | *Staphylococcus aureus* | *Staphylococcus aureus* |
| 13 | | Coxsackievirus A (CAV) |
| 14 | | Human rhinovirus C |
| 15 | | SARS-CoV-2 (2019-nCoV) |

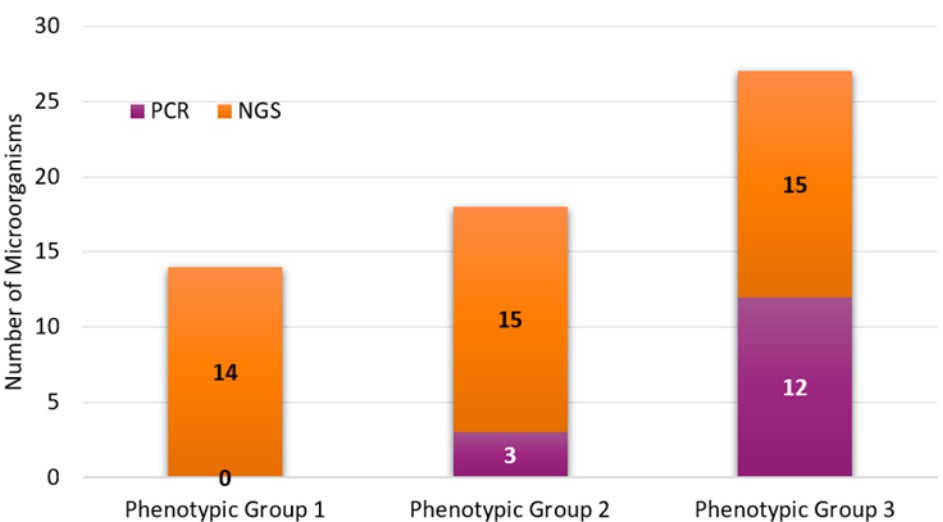

**Figure 2.** Comparative analysis of phenotypic grouping (PCR vs. mNGS; *n* = 29). To avoid any errors during position changes, please provide the combined image instead of editable pieces in the figure.

## 4. Discussion

Respiratory (co)infections remain a leading cause of global morbidity and mortality despite advances in diagnosis and treatment. Rapid identification and functional characterization of microorganisms is critical for the clinical management of respiratory disease. No doubt, improved diagnostics of complex upper respiratory (co)infections is key to understanding the trafficking potential to the lower respiratory tract and their pathogenic role in lower airway disease [25]. However, there are noted limitations with traditional syndromic approaches to pathogen detection (i.e., serology, microbiology, PCR) that can result in lag-time to diagnosis, misdiagnosis, or unsuitable treatment, which increased risk for trafficking and prolonged sickness [26]. This is important because the etiology of respiratory tract (co)infections goes undiagnosed in ~20 to 60% of patients with community-acquired pneumonia when using traditional syndromic laboratory testing approaches [14]. More robust mNGS hypothesis-free techniques, although yielding a broad microbiome, pose a taxonomic-to-clinically relevant challenge. From a laboratory point of view, these sequencing methods often require workflows that are technically challenging and tend to be siloed [27] because of the high capital investments needed for equipment, technology, and expertise—resulting in restricted capability for decentralized testing or application in resource-limited laboratories [14,28]. Yet from the clinical point of view, recognition of potential pathogenicity is crucial to estimating etiological relevance, optimizing treatment, and understanding outbreak conditions. Similarly, the recognition of broader pathogenic microorganisms can be pivotable to draw on for decentralized epidemiological layouts [29]. Assuredly, underdiagnosed upper respiratory (co)infections have potential for trafficking to the lower respiratory region and trigger adverse harm, especially in immunosuppressed hosts [30–34].

More recent advances have introduced a hybridization-capture-based mNGS approach with some initial success as a viable alternative for deciphering microbiota in clinical samples suspected of lower respiratory (co)infections [24]. However, exploring acute upper respiratory (co)infections with this approach to account for potential trafficking to the lower respiratory tract is less studied. Accordingly, we sought to test nasopharyngeal samples of patients suspected of upper respiratory (co)infection and (a) compare the findings of a commercially available respiratory PCR panel with those of a hybridization-capture-based mNGS respiratory panel and (b) phenotypically classify the findings for potential trafficking pathogenicity for the fields of laboratory and respiratory medicine to consider.

Second, the findings were characterized and phenotypically grouped using the automated IDbyDNA Explify® Platform (Illumina® Inc, San Diego, CA, USA) data analysis solution. Of the normal flora, colonizers, or contaminants noted in phenotypic group 1, most are innocuous, commonly existing in commensal relationships with their hosts, having rare or low trafficking potential for active disease. However, there are cases in the literature to be considered. For example, bacteria such as *H. haemolyticus* have potential for a rare invasive disease which can be overlooked and misidentified as *H. influenzae* [35]; *n. flavescens* has been identified to have a pathogenic role in immunocompromised and diabetic patients, and in rare cases has been linked to necrotizing pneumonia [36]; *D. pigrum* has been shown to trigger cases of nosocomial pneumonia [37]; *C. leadbetteri* is occasionally responsible for acute exacerbation of chronic obstructing pulmonary disease (COPD) [38]; *G. haemolysans* has been discovered to play a direct role in pulmonary exacerbations in patients with cystic fibrosis [39]; although considered to have lower virulence, *Ochrobactrum* spp. are beginning to emerge in the literature as a major opportunistic respiratory pathogen [40]; *P. melaninogenica* has been found to have an association with ventilator-associated pneumonia [41]; *R. mucilaginosa* has been linked to bronchiectasis [42]; and *C. pseudodiphtheriticum* has been associated with tracheobronchitis, pneumonia, and lung abscesses [43], while others have shown trafficking potential that includes various systemic infections, including meningitis and osteomyelitis [44]. Even more rare, but with clinical manifestation, is the fungus *A. alternata,* which has been shown to have opportunistic infectivity in the lower respiratory tract in patients with acquired immunodeficiency syndrome [45]. Im-

portantly, it is critical to distinguish the contaminations from the active infection during highly sensitive molecular diagnostics. Accordingly, we tested a no-template control (NTC) sample from end-to-end—sample processing and sequencing—and any organism detected in the NTC was considered contamination. Likewise, validation of the mNGS assay must establish the criteria to exclude potential contamination from clinical reporting. In phenotypic group 2 we detected microorganisms more likely to persuade lower respiratory tract infectivity—frequently associated with respiratory disease. First, we considered three bacteria detected in parallel by both assays, *M. catarrhalis* (*n* = 12/29), *H. influenzae* (*n* = 9/29), and *S. pneumoniae* (*n* = 9/29). Each of these bacteria has moderate to significant lower respiratory trafficking capacity. *M. catarrhalis,* although commonly noted as a cause of otitis media in children, has potential for COPD exacerbations and pathogenesis for bronchopulmonary infection post pulmonary aspiration [46,47]. Of course, *H. influenzae* and *S. pneumoniae* are common trafficking infectors originating in the upper respiratory tract. But even with vaccines, ~1000 people die annually of *H. influenzae* in the United States, whereas *S. pneumoniae* is the leading cause of global pneumonia mortality [48]—confirming the importance of differentiating these pathogens on any targeted laboratory testing application. Regarding the remaining 12 microorganisms characterized as phenotypic group 2 (10 bacteria, 1 virus, 1 fungus), the literature considers each associated with morbidity and mortality in immunocompromised and enervated individuals. Of more specific concern for trafficking potential is *H. parainfluenzae*, a commensal upper respiratory microorganism that with the acquisition of transmitted virulence can trigger a severe pathogen process in the lower respiratory tract [49]; *A. graevenitzii* as a cause of lung abscesses mimicking acute pulmonary coccidioidomycosis [50]; *A. xylosoxidans* for its predilection to worsen certain cases of cystic fibrosis [51]; and *S. mitis* and *S. maltophilia* due to their recognition as increasingly significant nosocomial pathogens that frequently exhibit multidrug resistance [52,53]. In addition to the identified bacteria, CMV spp. was found in 2/29 cases and has potential for severe symptomatic pneumonia in immunocompetent hosts [54], and *F. proliferatum* was observed in 1/29 cases, critical for its infectivity potential in lung transplant patients [55].

Phenotypic group 3 microorganisms are considered for their more significant pathogenicity aimed at respiratory disease. Importantly, 12/15 microorganisms were identified in parallel between PCR and mNGS—again confirming the significance of clinically relevant PCR targets in the laboratory. However, the hybridization-capture-based mNGS workflow discovered three pathogens not probed by PCR, each with significant infectivity potential for trafficking to the lower respiratory tract. First, SARS-CoV-2, the responsible agent for the COVID-19 pandemic with global deaths of 6.5 million and climbing [56]. The importance of including a probed target for SARS-CoV-2 in any respiratory case or laboratory test cannot be understated, and we leave this discussion to scholars who have written much on this topic [57–59]. We do, however, consider in more depth the additional two microorganisms exclusively identified by mNGS that were characterized in this group: CAV and HRV-C—both targeting bronchial and alveolar epithelial cells and explicit for excessive uncontrolled lung inflammation if not cleared. When considering these pathogens, it is important for clinicians to take into account most PCR panels, including the FTD® panel used in this study, use primers that do not distinguish between enterovirus and rhinovirus, and report positive amplification without any enterovirus/rhinovirus genus differentiation. While targeted PCR has tremendous advantages that include cost, scale, speed, sensitivity and specificity, the results are restricted in divergence, diversity, genotype, and functional potential, ensuing in restricted clinical significance in many cases of respiratory infection [60,61].

The genus *Enterovirus* has been divided into a total of 12 species; enteroviruses A-J (which include the coxsackievirus, poliovirus, and echovirus subspecies) and Human rhinoviruses A-C. Coxsackie viruses— reactivating pathogens in immunosuppressed patients—are divided into two major serotype groups: group A and group B. Although type A coxsackieviruses cause herpangina (a common childhood illness), acute hemorrhagic conjunctivitis, and hand-foot-and-mouth disease, there are case reports—from toddler

to adult—of fatal pulmonary illnesses as a result of CAV [62,63]. Of the three Human rhinoviruses A-C, HRV-C, identified in 2006, is more frequently associated with its severity of clinical manifestations for lower respiratory tract disease and severe illness [64,65]. For this reason, the hybridization-capture-based mNGS workflow may have clinical significance for its ability to differentiate these viruses at the species level—especially valuable in pediatric and immunocompromised cases [66,67]. Compositions of upper respiratory tract (URT) microbial communities is primarily determined by in combination of the location within the URT and external factors, such as ageing, diseases, immune responses, olfactory function, and lifestyle habits such as smoking [68,69]. Therefore, further investigations for understanding the lower respiratory track microbial composition in diverse populations by gender, age, immune status, and comorbidities are warranted.

*Implications for Respiratory Medicine*

Because lower respiratory (co)infections are a communal source of worldwide morbidity and mortality [1], there is a need to identify preceding or concurrent upper respiratory (co)infections that may trigger harmful trafficking effects on lower respiratory disease. And because the clinical picture of lower respiratory (co)infections can be complex and heterogenous with multiple etiological agents (i.e., bacteria, fungi, viruses, and parasites), rapid and accurate diagnosis can reduce the risk of protracted (co)infections and advanced application of pathogen-specific medication. However, when considering current syndromic approaches to diagnosis, respiratory medicine must consider limitations in laboratory testing rooted in techniques based upon a priori assumptions. In such cases, for accurate identification of diagnosis of fastidious microorganisms, rare and atypical pathogens, or inviable agents post antimicrobial therapy, a hybridization-capture-based mNGS approach may offer a viable alternative.

This is because clinical samples often comprise nucleic acids derived from the host (human), whereas the nucleic acids of interest for a hybridization-capture-based mNGS approach are microbial (or nonhuman). Thus, hybridization-capture-based enrichment can secure relevant microbiota nucleic acid instead of depleting abundant human nucleic acid (genomic DNA, ribosomal or mitochondrial RNA). Consequently, this hybridized approach can provide clinically relevant information from more cost-effective and lower-throughput sequencing techniques, important for decentralized testing and laboratories with limited resources. Although researchers have demonstrated that targeted hybridization-capture-based mNGS workflows for lower respiratory pathogen-specific sequences have a critical implication [24], preceding or concurrent consideration of upper respiratory pathogenic infectivity and the potential trafficking effects on lower respiratory disease are less studied with this approach.

This is an important distinction from other amplicon sequencing methods laboratorians commonly use to analyze target-specific genomic regions of respiratory microbiota—NGS of the 16S rRNA gene and target-agnostic mNGS. However, NGS of the 16S rRNA gene is limited to bacterial identification and is often restricted to the genus level [70]. Furthermore, the 16S rRNA-based approach does not provide crucial AMR marker detection for focused medication therapy [71]. Alternatively, the target-agnostic mNGS approach is becoming an increasingly viable technique for obtaining microbial nucleic acid sequence information for the diagnoses and treatment of pulmonary (co)infections [72]. However, this method is based on the depletion of host DNA, and deep sequencing of the remaining DNA requires a higher depth because of a high degree of residual host DNA—requiring high capital investment (~1 million USD) in technology, bioinformatic specialization, and potential for turnaround lags because sample pooling is often needed to reduce run costs. As such, target-agnostic mNGS is most suitable in research laboratories or for epidemiological surveillance (not unlike the recent COVID-19 pandemic). Thereupon, these amplicon sequencing methods are largely unsuitable for deployment for routine diagnostic use, especially in decentralized or resource-limited clinical laboratories [29].

The ability of mNGS to outperform common diagnostic procedures in detecting unidentified pathogens has been reported. For example, [73] analyzed pathogens of respiratory infection by mNGS in bronchoalveolar lavage fluid (BALF) from immunocompetent pediatric patients with respiratory failure. The study reported significant identification of bacterial or viral sequencing reads in 8/10 patients not identified by conventional methods. Miao et al. [74] demonstrated that mNGS yielded higher specificity and sensitivity than microbial culture when identifying *Mycobacterium tuberculosis*, viruses, anaerobes, and fungi. Recent studies have reported more than 80% sensitivity of mNGS in detecting respiratory pathogens compared to that of traditional methods [75]. Moreover, co-infection is very common in clinical settings, with studies reporting up to 70% of patients co-infected with bacteria–viruses, bacteria–fungi, or virus–fungi—with mNGS successfully identifying all co-infections compared to PCR analysis [76]. Likewise, our current study identified co-infection in (26/29) samples using the hybridization-capture-based approach. This favors the advantage of using hybridization capture to identify a wide range of microorganisms in a single analysis. Overall, the hybridization-capture-based mNGS workflow appears to be an emerging and promising technology for detecting respiratory pathogens more effectively and with more clinical relevance than conventional culture or PCR analysis. In one patient sample, *M. catarrhalis* was detected by PCR (13/29) but not by mNGS (12/29). This discrepancy can be attributed to the competitive disadvantage of hybridization-capture-based shallow (0.5 M reads) sequencing over targeted PCR. Higher depth of coverage can overcome this challenge but at a higher cost. Alternatively, targeted PCR can also compensate for some limitations of the mNGS approach [77]. A negative mNGS result does not necessarily indicate that the patient was not clinically infected and, therefore, clinical features are equally important, particularly in complicated infection cases [78]. Technologically, very limited discrepancies are reported between the PCR and mNGS regardless of the organism type if the probes for the organisms are included in the targeted mNGS panel [79]. Similarly, mouse colony microbiome analysis has also suggested a high degree of concordance between the technologies [80]. Therefore, when considering strategic deployment of these technologies, a thorough understanding of the clinical demands must be coupled with any added financial burden needed to increase the capacity of pathogen discovery.

Nevertheless, differentiating pathogens from commensal and colonized microorganisms is very challenging in respiratory (co)infections. Lungs in healthy and diseased individuals host different bacterial strains, and asymptomatic or potentially pathogenic organisms are ubiquitously present in the lungs. For example, 20–50% of healthy airways are colonized by opportunistic bacteria such as *S. pneumoniae* and *H. influenza*—both considered to have high trafficking and lower respiratory infectivity potential. Although phenotypic grouping can aid the characterization of pathogenicity, other clinical factors should be considered for accurate diagnosis based upon data generated from a hybridization-capture-based mNGS workflow—especially immunocompromised and immunosuppressed patients who are more vulnerable to (co)infections caused by a wide range of common and uncommon pathogens.

Finally, even though target-agnostic sequencing can identify the entire microbiome without a priori knowledge, the targeted hybridization-capture-based mNGS workflow is likely to have better application in clinical settings. This is because of the precise detection of clinically relevant organisms and better adaptability on low-throughput sequencing instruments. For example, using the hybridization-capture-based mNGS workflow, we correctly identified organisms in NATtrol™ Respiratory Panel 2.1 controls (ZeptoMetrix, Buffalo, NY, USA) in five different runs from only 0.5 million reads/sample. The lowest total reads ($\geq$0.5 M/sample), and a combination of median depth ($\geq$1x), coverage ($\geq$40%), and RPKM ($\geq$10), resulting in over 90% accurate detection of the control organisms, were accepted as a cut-off for unknown targets. Clinical laboratories must re-establish these criteria before deploying the mNGS test in clinical testing as a laboratory-developed test. This approach can essentially analyze ~24 samples with a small benchtop MiniSeq™ System, making it more easily adaptable and cost-effective for the clinical setting. Importantly, there

are no current Federal-Drug-Administration-approved devices or kits for sequencing infectious disease, requiring, in the United States, the extensive validation of the test's accuracy, sensitivity, and specificity according to Clinical Laboratory Improvement Amendments (CLIA) guidelines (https://www.cms.gov/regulations-and-guidance/legislation/clia accessed on 1 March 2022). However, clinical decisions should be made after comprehensive consideration of all the available clinical and diagnostic information.

## 5. Conclusions

Much of respiratory medicine is reliant on timely and precise diagnostics for critical treatment. However, respiratory infections remain a leading cause of global mortality and morbidity despite advances in diagnosis and treatment. In this study, we (a) compared nasopharyngeal samples from patients suspected of acute upper respiratory infection between a commercially available PCR assay and a targeted hybridization-capture-based mNGS workflow and (b) demonstrated that the hybridized approach may provide tremendous advantage in deciphering the etiological agent of respiratory (co)infections and provide clinical relevance for trafficking potential. This is important because the trafficking potential from the upper to the lower respiratory tract and infection severity depend on pathogen virulence, concomitant infections, and underlying respiratory comorbidities [81].

This is significant to respiratory medicine because this technology can be used to supplement current syndromic-based tests, and data can quickly and effectively be phenotypically characterized for clinical (co)infection and comorbid consideration. This has significance for laboratory medicine because it demonstrated that this approach can rapidly be interpreted with a user-friendly and reliable platform for collective intention without overburdening laboratory investments in technology and people [82,83]. Furthermore, this approach could be advanced into pan-microbial diagnostic testing that utilizes a single workflow for all specimen types. Although we have demonstrated the analytical advantage of a targeted hybridization-capture-based mNGS workflow over targeted PCR analysis, further investigations are required to establish the clinical relevance of phenotypic classifications and their value to trafficking predispositions and utility in respiratory medicine.

**Supplementary Materials:** The following supporting information can be downloaded at: https://www.mdpi.com/article/10.3390/arm91010006/s1, Table S1: RAW; Table S2: Targets.

**Author Contributions:** Conceptualization, R.E.C. and R.S.; methodology, S.A. and R.S.; validation, S.A., C.R., and R.S.; data curation, S.A., C.R., A.S., and V.K.T.; writing—original draft preparation, S.A. and R.S.; writing—review and editing, R.E.C.; supervision, R.S.; project administration, R.S.; All authors have read and agreed to the published version of the manuscript.

**Funding:** This research received no external funding.

**Institutional Review Board Statement:** This research used de-identified samples and institutional IRB exempted the study.

**Informed Consent Statement:** Patient consent was not applicable due to research conducted on de-identified samples.

**Data Availability Statement:** The Bioinformatics pipeline (Explify®) used to accumulate data did not generate sample-specific fastq files; thus, data were not deposited in any public database. mNGS-run BLC files and demultiplexing barcode information are available upon reasonable request.

**Conflicts of Interest:** The authors declare no conflict of interest.

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
