# Peer review of "Deciphering Microbiota of Acute Upper Respiratory Infections: A Comparative Analysis of PCR and mNGS Methods for Lower Respiratory Trafficking Potential"

_arm, doi:10.3390/arm91010006_

Round 1
Reviewer 1 Report
Authors try to use NGS and PCR based methods to identify the diversity of the pathogens (both bacterial and viral in the respiratory tract). The topic is original and relevant to diagnostics. However I have some suggestion sor concerns, which might enhance the paper when addressed by the authors. Please find them below:
1. How can immunocompetent patients be better managed by identification through mNGS method? Explain a bit in discussion section.
2. Authors mention ‘orchestrate specific leukocyte trafficking molecules’, yet no evidence or literature support provided. Please edit this.
3. How robust are mNGS results and what is the chance of false positives?
4. M. catarrhalis was detected by PCR and not mNGS. Then how can we be sure that mNGS will always work if this assay is introduced in clinics? Any alternate strategy or future implementable perspective?
5. How do you correlate your findings to that of Jeong et al. (2022). DOI: 10.3390/microorganisms10020324. They suggest that approach to quantifying specific microorganisms by applying the qRT-PCR method can compensate for the concerns (potential issues) of NGS.
6. There was least discrepancy between NGS and PCR method for visruses, is this supported by literature? In mice model?
7. It could be interesting to see diversity in the respiratory tract fractions like lung, nasal and oral, in the future. Reference: 10.1371/journal.pone.0222589. The findings could help validate if the disparity was just in one area or all. Any other subset e.g. population or gender based variation could be interesting as well. This could be added in future perspectives.
8. Conclusions are consistent with findings but authors need to incorporate some more literature that has been published recently on the topic and missed in the manuscript.
Reviewer 2 Report
The article “Deciphering microbiota of acute upper respiratory infections: a comparative analysis of PCR and mNGS methods for lower respiratory trafficking potential” compares two technics to identify respiratory infections agents. It is a very important study that highlights the advantages of using an mNGS approach, covering many bacterial and virus species. Considering the broad spectrum of respiratory pathogen agents with an outbreak potential, the use of a diagnostic technic is important to support the clinics and hospitals. Although it is an important study topic, the manuscript needs extensive review and some suggestions are listed below.
Major
The figures 1 and 2 need to be revised in terms of aesthetics and quality. Figure 1 needs to clear the background. The figure 2 chart does not need to be in 3D, just a simple bar graph will do. Include the overlapping of PCR and mNGS, and indicate what is in the y-axis.
Why the AMR analysis was not included in this study? Explain in the methods section.
The PCR-based technic did not include SARS-CoV-2 in the panel. The author only include one sentence in the discussion and did not address this. The reason for this limitation should be discussed. Was the panel developed before the SARS-CoV-2 pandemic? Do they have an updated version or recommendations?
Contamination is a big problem when working with mNGS, especially for diagnosis purpose. What were the controls e strategies used to evaluate this and identify contaminants when and if it happens?
Why use a median depth of 1X. Is this appropriate for the mNGS analysis? Discuss this.
There are some tables cited in the text but are not in the paper. Review and correct this.
Minor
Include the period (dates) of the sample collection in the methods section.
The RPIP preconizes that each sample should have 1 million reads and you included samples with 0.5 million reads. I agree that 0.5 million reads per sample in an RPIP assay is acceptable, but this should be discussed by exploring the scientific basis and reason to include them.
Round 2
Reviewer 1 Report
Authors have now responded to all my points. Manuscript is now fine for publication.
Author Response
Thank you for your productive comments to our manuscript.
Reviewer 2 Report
All the major and minor suggestions were made. No new comment.
Author Response

(The authors gave the same response as above.)
